# In vivo detection of optically-evoked opioid peptide release

**Ream Al-Hasani[1,2,3†]\*, Jenny-Marie T Wong[4†], Omar S Mabrouk[4,5], Jordan G McCall[1,2,3,6], Gavin P Schmitz[1,3], Kirsten A Porter-Stransky[7], Brandon J Aragona[7], Robert T Kennedy[4,5‡]\*, Michael R Bruchas[1,6,8,9‡]\***

[1]Department of Anesthesiology, Division of Basic Research, Washington University School of Medicine, St. Louis, United States; [2]Department of Pharmaceutical and Administrative Sciences, St. Louis College of Pharmacy, St. Louis, United States; [3]Center for Clinical Pharmacology, Washington University School of Medicine and St. Louis College of Pharmacy, St. Louis, United States; [4]Department of Chemistry, University of Michigan, Ann Arbor, United States; [5]Department of Pharmacology, University of Michigan, Ann Arbor, United States; [6]Washington University Pain Center, Washington University School of Medicine, St. Louis, United States; [7]Department of Psychology, University of Michigan, Ann Arbor, United States; [8]Department of Neuroscience, Washington University School of Medicine, St. Louis, United States; [9]Department of Anesthesiology and Pain Medicine, Center for the Neurobiology of Addiction, Pain, and Emotion, University of Washington, Washington, United States

**\*For correspondence:**
al-hasanir@wustl.edu (RA-H);
rtkenn@umich.edu (RTK);
bruchasm@wustl.edu (MRB)

†These authors contributed equally to this work
‡These authors also contributed equally to this work

**Abstract** Though the last decade has seen accelerated advances in techniques and technologies to perturb neuronal circuitry in the brain, we are still poorly equipped to adequately dissect endogenous peptide release in vivo. To this end we developed a system that combines in vivo optogenetics with microdialysis and a highly sensitive mass spectrometry-based assay to measure opioid peptide release in freely moving rodents.
DOI: https://doi.org/10.7554/eLife.36520.001

## Introduction

Neuropeptides are the largest class of signaling molecules in the central nervous system where they act as neurotransmitters, neuromodulators, and hormones (*Wotjak et al., 2008*). In vivo neuropeptide detection is technically challenging as neuropeptides can be rapidly cleaved by peptidases and undergo posttranslational modifications. To further complicate detection, neuropeptides are found at orders of magnitude lower concentrations compared to classical neurotransmitters (i.e. glutamate, GABA, and the biogenic amines) and adsorb to a variety of surfaces during sample handling (*Zhou et al., 2015*; *Maes et al., 2014*).

For decades, neuropeptides have been assayed with immunoaffinity-based techniques due to their high sensitivity (*Maidment et al., 1989*; *Maidment et al., 1991*). However, selectivity remains a concern due to poor antibody specificity for full length vs. truncated peptides that contain identical binding epitopes. Nanoflow liquid chromatography-mass spectrometry (nLC-MS) is a powerful alternative for peptide detection because it provides high sensitivity and specificity without the need for an immunocapture step. To this end, nLC-MS has been used successfully to detect a number of endogenous peptides derived from intracranial sampling techniques such as microdialysis (*Van Wanseele et al., 2016*).

Opioid peptides are prominent neuromodulators for regulating motivated behaviors. Though understanding their endogenous release properties is critical for dissecting the neural circuits that mediate these behaviors it has been extremely difficult to reliably achieve thus far. The reason primarily being that opioid peptides are very similar in structure and origin making selective detection of peptides and their subtype fragments virtually impossible. All opioid peptides, except nociceptin share a common N-terminal Tyr-GlyGly-Phe signature sequence, which interacts with opioid receptors. The preproenkephalin gene encodes several copies of the pentapeptide Met-enkephalin and one copy of Leu-enkephalin. The preprodynorphin gene also encodes multiple opioid peptides, including dynorphin A, dynorphin B, and neo-endorphin. The preprodynorphin-derived peptides all contain the Leu-enkephalin N-terminal sequence. To further complicate things the enkephalins preferentially bind to the δ over μ opioid receptor whereas dynorphin preferentially binds to the κ opioid receptor.

Despite these complexities some studies have reliably detected opioid peptides in rat models (*DiFeliceantonio et al., 2012*; *Mabrouk et al., 2011*; *Lam and Gianoulakis, 2011a*; *Lam and Gianoulakis, 2011b*; *Lam et al., 2008*; *Lam et al., 2010*), however, it has not been previously possible to reliably detect evoked neuropeptide release in a cell-type selective manner using transgenic mouse models. However, with the advent of optogenetic approaches allowing researchers to spatially target and manipulate neuronal firing patterns the specific neuronal properties of neuropeptide release can now more likely be empirically determined.

We and others have recently shown that discrete targeting of the kappa opioid system in the nucleus accumbens (NAc) can modulate both rewarding and aversive behaviors (*Al-Hasani et al., 2015*; *Castro and Berridge, 2014*). We showed that photostimulation of dynorphin (dyn) cells in the ventral nucleus accumbens shell (vNAcSh) elicited robust aversive behavior, while photostimulation of dorsal NAcSh dyn (dNAcSh) cells induced a place preference that was positively reinforcing, however, no successful measurements of optogenetically-evoked neuropeptide release in vivo have been reported to date. Furthermore, few reports have extensively investigated regional distinctions within the NAc shell and no reports to our knowledge have successfully measured optogenetically-evoked neuropeptide release in vivo.

To quantify evoked-neuropeptide release in anatomically and behaviorally distinct regions, we pursued a method to reliably detect neuropeptides in the nucleus accumbens during photostimulation of opioid-containing neurons. We developed a custom optogenetic-microdialysis (opto-dialysis) probe that simultaneously provides photostimulation with the ability to sample local neurochemical release in awake, freely moving mice. We established a targeted nLC-MS method to analyze the opioid peptides dyn (fragment dynorphin $A_{1-8}$) and enkephalins (leu- and met-), as well as dopamine, GABA and glutamate. We applied these techniques to precisely investigate regional differences between the vNAcSh and dNAcSh neuropeptide release dynamics. This system allows quantification of neuropeptide release while directly controlling cell-type selective neuronal firing in the NAcSh.

## Results

To establish a quantitative assay, we used a custom synthesized isotopically labeled dyn (DYN*, YGGFLRRI with isotope $^{13}C_6$$^{15}N_1$-leucine) with a mass shift of +7 and a +3.5 *m/z* shift (+2 charge state) as an internal standard (IS) to account for variability during the LC injection, surface adsorption, and ionization efficiency. High concentration injections of DYN* were fully resolved by the mass spectrometer and did not result in cross talk with endogenous dyn (*Figure 1a and b*), demonstrating that the addition of labeled DYN* does not contribute to or interfere with the endogenous dyn signal, while maintaining a similar retention time. To further validate the use of DYN* as an internal standard when detecting opioid peptides we prepared a linear calibration curve with a mixture of dyn, Leu-Enkephalin (LE) and Met-Enkephalin (ME) standards spiked with DYN* and detected all four compounds within 5 min following loading and desalting the sample on the column. Four distinct peaks are shown in the reconstructed ion chromatographic trace, confirming the separation and reliable detection of all three opioid peptides with isotopically labeled DYN* in one sample (*Figure 1c*). The addition of the DYN* isotope to our assay further improves quantification by improving relative standard deviation for repeated injections of standards (*Figure 1d–f*). Ratios of dyn, LE, and ME to a consistent DYN* were used for calibrations and analysis of standards for

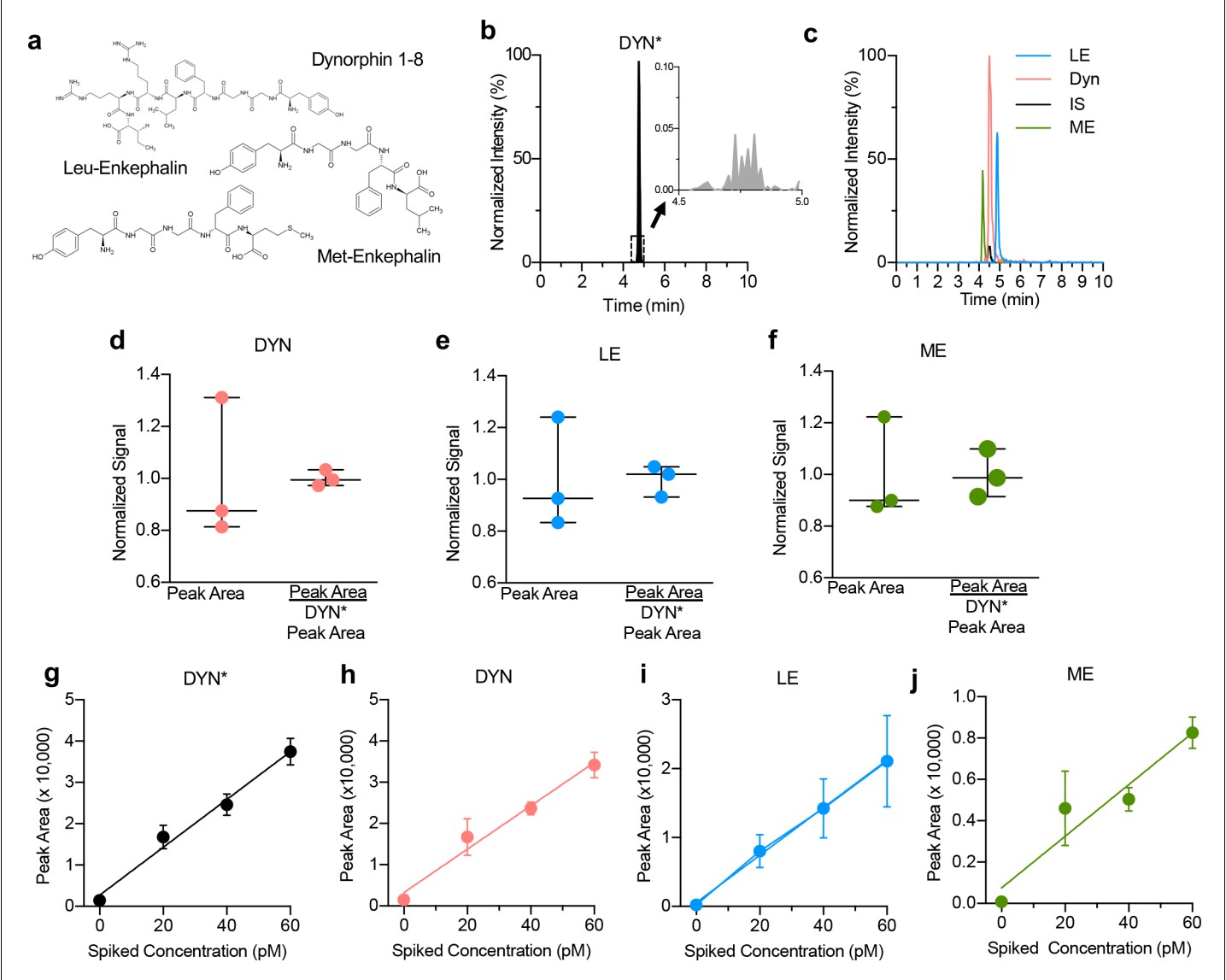

**Figure 1.** Optimization of opioid peptide detection parameters. (**a**) Chemical structures of Dynorphin A 1–8, Met-Enkephalin and Leu-Enkephalin (**b**) Isotopically labeled dynorphin as an internal standard for quantitative analysis. High concentration injections of DYN* did not show significant traces of endogenous dyn (inset trace). A 500 pM sample of DYN* was injected while monitoring both the endogenous dyn (491 → 435 $m/z$) and isotopically labeled DYN* (495 → 438 $m/z$) mass-to-charge transitions. (**c**) Nano LC-MS chromatograms of 100 pM standards. Reconstructed ion chromatogram of ME, dyn, DYN*, and LE. (**d**) Addition of isotopically labeled DYN* results in better quantification of dyn A$_{1-8}$, (**e**) Leu-Enkephalin and (**f**) Met-Enkephalin. (**g–j**) No effect of ionization suppression from the matrix, as shown for DYN*, Dyn A$_{1-8}$, Leu-Enkephalin and Met-Enkephalin, respectively. Bulk dialysate was collected and spiked with known amounts of standard. This resulted in a linear response that corresponded with the signal increase from the original sample analyte plus the additional analyte, showing no effect of ionization suppression from the matrix. Four replicates per sample; data shown as average ± SD.

DOI: https://doi.org/10.7554/eLife.36520.002

quantitative analysis for all experiments shown here. No effect of dialysate matrix for any targeted analyte was observed (*Figure 1g–j*).

For in vivo detection in freely moving mice, we developed a customizable microdialysis probe (i.e. customizable length, depth and sampling area) with an integrated fiber optic to locally sample proximal to the site of photostimulation in the brain (*Figure 2a*, *Figure 2—figure supplemenrt 1a–e*). Traditional dialysis probes incorporate inlet and outlet tubing encased in a semi-permeable membrane enclosed with epoxy, and further encased in a stiff cannula for rigidity and robustness. To

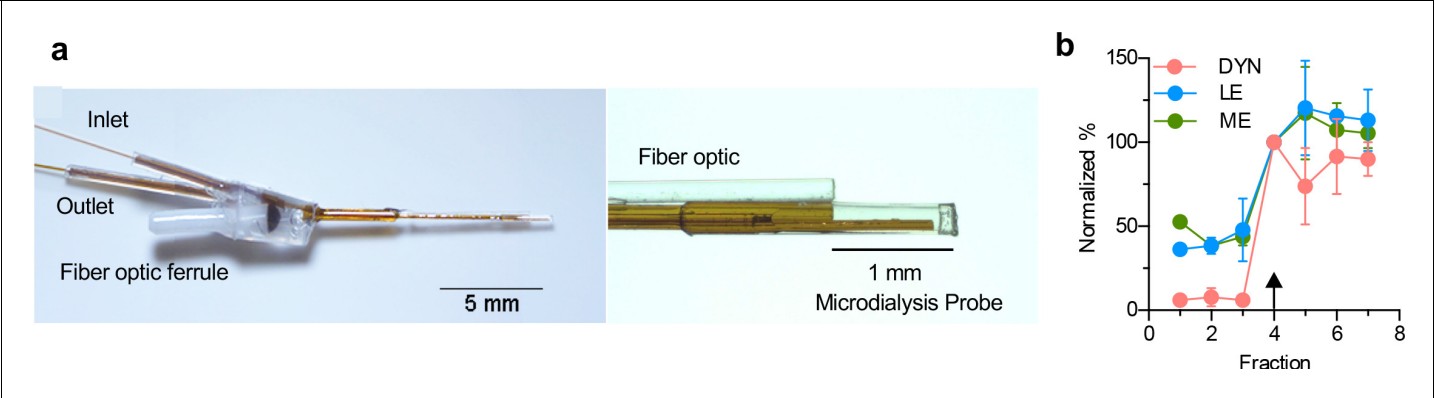

**Figure 2.** Design and function of opto-dialysis probe. (**a**) Images of the optodialysis probe. (**b**) Trace of an in vitro step change from 100 pM of DYN, LE, and ME stock solution to a solution of 1 nM Dyn and 400 pM LE and ME. The arrow indicates the first fraction in which the peptide was expected to change. Data were normalized to fraction 4, the fraction expected to reflect elevated concentration stock change. Data shown as average ± SD, n = 4 probes.

DOI: https://doi.org/10.7554/eLife.36520.003

The following figure supplement is available for figure 2:

**Figure supplement 1.** Step-by-step illustration of the custom-made integrated optogenetic-dialysis probes measure peptides and small molecules in freely moving animals.

DOI: https://doi.org/10.7554/eLife.36520.004

minimize the size of the opto-dialysis probe we did not include the final external casing and instead took advantage of the natural rigidity of the fiber optic to support the dialysis inlet-outlet assembly (*Figure 2a*, *Figure 2—figure supplemenrt 1a–e*). This resulted in a maximal probe diameter of 480 µm. To maximize concentrations of peptides entering the probe, we used a 60 kDa molecular weight cutoff, polyacrylonitrile membrane with a slight negative charge for optimal peptide recovery (AN69, Hospal, Bologna Italy) (*Zhou et al., 2015*; *Maidment et al., 1989*) and showed we can measure changes in peptide stock concentration within the 15 min fraction collection time (*Figure 2b*).

To evoke and measure in vivo peptide release we injected AAV5-EF1α-DIO-ChR2-eYFP in to either the vNAcSh or the dNAcSh of preprodynorphin-IRES-cre (dyn-cre) mice and implanted the custom opto-dialysis probes 3 weeks later (*Figure 3a*). Following recovery artificial cerebrospinal fluid (aCSF) was perfused through the device at 0.8 µL/min and fractions were collected on ice every 15 min, generating 12 µL volumes, 2 µL of which were aliquoted and used for a small molecule detection assay. Three baseline fractions were collected prior to a fraction capturing 15 min of 10 Hz (10 ms pulse width) photostimulation, and six additional fractions were collected following photostimulation. As a positive control for detecting dynamic changes in vivo by nLC-MS and to establish sampled neuron responsivity to stimuli, we infused 100 mM K$^+$ aCSF at the end of each collection experiment. The influx of K$^+$ ions causes depolarization, resulting in vesicular exocytosis, which is expected to evoke a large increase in peptide concentration. Mice were included in the study if 100 mM K$^+$ stimulation resulted in a positive increase in analytes at the end of the experiment (*Figure 3—figure supplement 2a–c*) and if correct anatomical probe placement and viral expression were confirmed (*Figure 3—figure supplement 3a and b*). In the current study 81.1% (15/18) of the mice were included in the analysis.

A significant increase in dyn was detected in dyn-cre positive mice during photostimulation in both the vNAcSh (interaction effect; t = 3.941, p<0.001) and dNAcSh (interaction effect; t = 3.012, p=0.003), compared to control mice (*Figure 3b*). Interestingly, in the vNAcSh there was also a sustained increase in dyn after photostimulation (interaction effect; t = 2.499, p=0.014) (*Figure 3b*). Dyn release during photostimulation was also significantly higher in the vNAcSh compared to the dNAcSh (interaction effect; t = 2.749, p=0.007) and post stimulation (interaction effect; t = 2.806, p=0.006) (*Figure 3b*). Photostimulation of vNAcSh dyn neurons was previously shown to cause aversive behavior, consistent with early pharmacological studies which linked dyn with negative emotional states (*Al-Hasani et al., 2015*; *Bals-Kubik et al., 1993*). Here, we demonstrate sufficiently

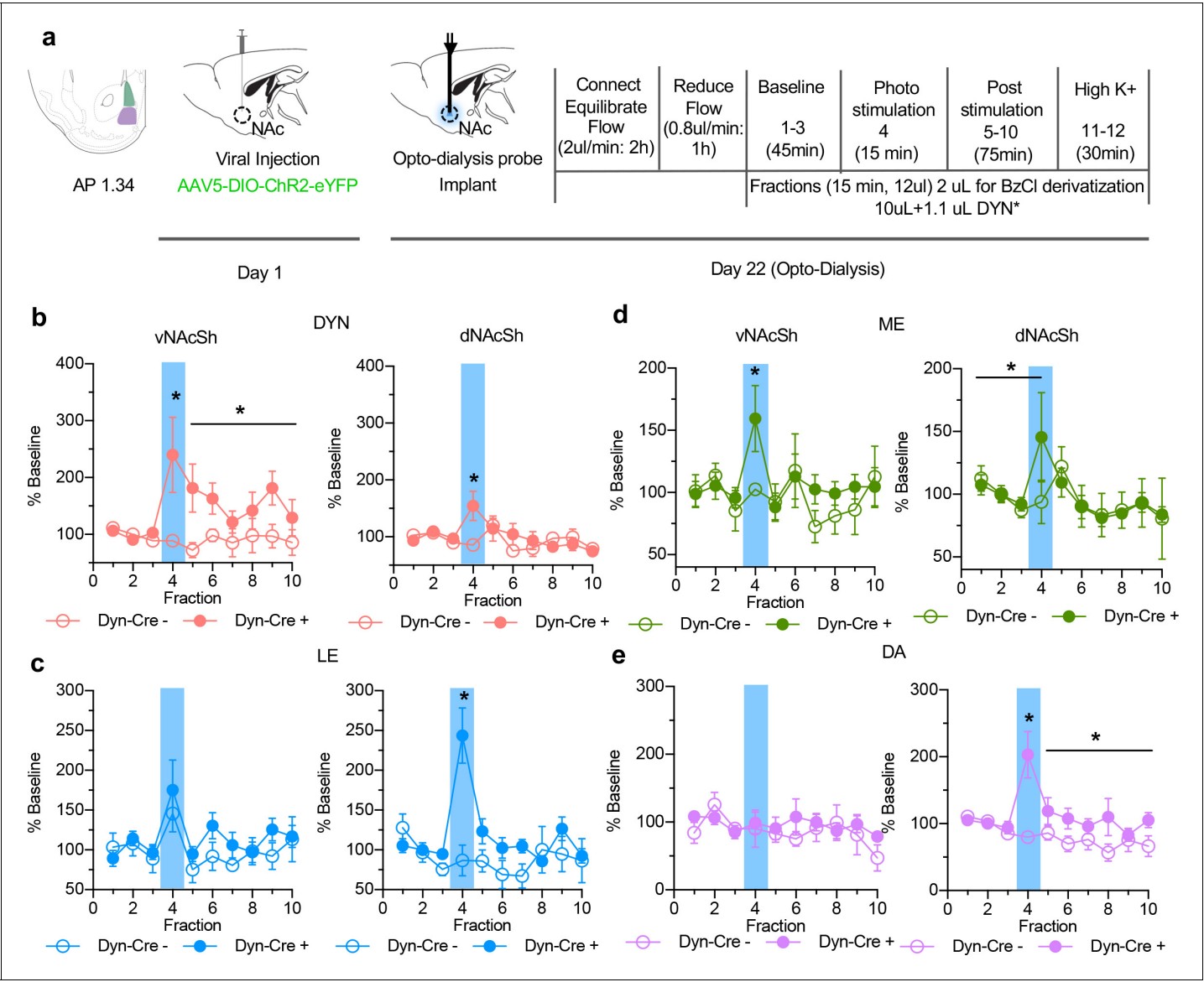

**Figure 3.** In vivo changes in opioid peptide release following photostimulation of dynorphin cells. (**a**) Timeline of experimental procedure outlining viral injection, probe implantation and dialysate collection. (**b**) Extracellular opioid peptide release shown as % baseline in vNAcSh $dyn_{1-8}$, (left panel, n = 8), dNAcSh $dyn_{1-8}$ (right panel, n = 6). (**c**) vNAcSh Leu-Enkephalin (left panel, n = 4), dNAcSh Leu-Enkephalin (right panel, n = 6). (**d**) vNAcSh Met-Enkephalin (left panel, n = 7) and dNAcSh Met-Enkephalin (right panel, n = 7). (**e**) Small molecules simultaneously collected and shown as % baseline in vNAcSh Dopamine (left panel, n = 7), dNAcSh Dopamine (right panel, n = 7).

DOI: https://doi.org/10.7554/eLife.36520.005

The following figure supplements are available for figure 3:

**Figure supplement 1.** In vivo K + depolarization.

DOI: https://doi.org/10.7554/eLife.36520.006

**Figure supplement 2.** Hits maps showing placement of opto-dialysis probes (**a**) vNAcSh, closed purple circles represent correct hits, open circles represent misses.

DOI: https://doi.org/10.7554/eLife.36520.007

**Figure supplement 3.** Small molecules simultaneously collected and shown as % baseline in (**a**) vNAcSh GABA (left panel, n = 6), dNAcSh GABA (right panel, n = 6).

DOI: https://doi.org/10.7554/eLife.36520.008

discrete dynorphin detection to measure different levels of peptide in two regions of the NAc shell separated by 1 mm.

We simultaneously detected robust release of LE and ME in both the vNAcSh and dNAcSh. During photostimulation of dyn-containing cells in the dNAcSh, LE levels were significantly elevated compared to controls (interaction effect; t = 5.384, p<0.0001) (*Figure 3c*). In contrast, no changes in LE were detected during photostimulation of dyn-containing cells in the vNAcSh (*Figure 3c*). Converse to this, we observed a significant increase in ME during photostimulation of dyn-containing cells in the vNAcSh (comparing cre + and cre-, interaction effect; t = 2.824, p=0.006). However, the same effect was not significant in the dNAcSh (comparing cre + and cre-, t = 1.78, p=0.053) (*Figure 3d*). Importantly, we observed a significant change in ME in the dNAcSh when compared to its baseline (t = 1.94, p=0.033), and there was no significant difference between the effect of photostimulation in ventral and dorsal (i.e. both are increased). However, the lack of a significance between Cre + and Cre- in dNAcSh is likely due, at least in part, to the fact the baseline levels of opioid peptides in dNAcSh are higher than the vNAcSh (7.114 pM versus 2.71 pM, respectively). Though data are represented as % of baseline the range of absolute dialysate concentration detected for dyn was 0.28–0.44 pM in dNAcSh and 0.13–0.28 pM in vNAcSh; LE 1.39–3.28 pM in dNAcSh and 1.30–2.24 pM in vNAcSh; ME 6.19–8.69 pM in dNAcSh and 2.57–4.11 pM.

To determine if peptide release corresponded with small molecule neurotransmitter release, we applied a benzoyl chloride (BzCl) derivatization LC-MS method to monitor three small molecules, dopamine, GABA and glutamate in dialysis samples (*Song et al., 2012*; *Wong et al., 2016*). Interestingly, dopamine increased in the dNAcSh during photostimulation (interaction effect; t = 5.007, p<0.0001), which persisted following stimulation (interaction effect; t = 2.081, p=0.039) (*Figure 3e*). Levels of GABA also increased in the vNAcSh following photostimulation (t = 2.363, p=0.020) and had a prolonged response (t = 4.744, p<0.0001) (*Figure 3—figure supplement 3a*). There were no significant changes in glutamate levels during photostimulation in either vNAcSh or dNAcSh (*Figure 3—figure supplement 3b*).

## Discussion

The data show that photostimulation of dyn-containing cells in the vNAcSh and dNAcSh results in a detectable increase in dyn release, which is greater in the vNAcSh. This result correlates with previous behavioral data demonstrating that dyn release likely drives a region dependent preference and aversion behavior (*Al-Hasani et al., 2015*). We observed a significant increase in LE and dopamine during photostimulation in the dNAcSh. Importantly, we previously observed a preference behavior when photostimulating dyn cells in the dNAcSh and these data suggest that the preference may be related, at least in part, to the detected increases in dopamine and LE. Furthermore, many studies have shown that levels of both dopamine and LE increase during preference or reward (*Olds, 1982*; *Spanagel et al., 1990*; *Bruijnzeel, 2009*). Importantly, using this method it is not possible to know whether LE is derived from pDyn or pENK but in future experiments we can combine this with conditional knockout approaches to selectively delete the peptides in specific cell types, so that we can later determine what happens to these signals in the absence of expression of particular peptides in D1 vs D2 cells and subregions. In addition, we cannot preclude interactions between MSN cell types, specifically that optogenetic activation of D1 cells could impart changes on D2 cell activity thereby releasing Met-Enk, indirectly, and/or via indirect circuits outside the NAc. In addition, dopamine release via D1 MSN stimulation, could be via two mechanisms 1) via indirect suppression of D1 projections back to VTA GABA neurons (*Edwards et al., 2017*), which acts via GABA-B to disinhibit VTA DA neurons. This would then result in dopamine release in the NAc following D1 stimulation. Furthermore, an alternative, presynaptic regulation of DA release from VTA to NAc projections is also plausible, and recently been investigated by multiple groups (*Lemos et al., 2016*; *Dobbs et al., 2016*).

The observed changes in GABA in both vNAcSh and dNAcSh are as predicted due to the fact that dyn-containing medium spiny neurons are GABAergic. The observed increase in ME is intriguing, as ME is not released from the same cells as dynorphin. There has been some evidence to suggest that enkephalins play a role in regulating disruptions in homeostasis following chronic exposure to drugs of abuse (*Kreek and Koob, 1998*; *Shoblock and Maidment, 2007*). Specifically, increases in enkephalin release during drug withdrawal has been shown through peptide tissue content,

mRNA levels and microdialysis of extracellular enkephalins in a number of brain regions including the nucleus accumbens (*Nylander et al., 1995*). Concurrently, studies have measured increases in dynorphin expression during withdrawal (*Isola et al., 2008*; *McCarthy et al., 2010*; *Przewłocka et al., 1997*; *Rylkova et al., 2009*; *Lindholm et al., 2000*). It seems possible therefore that when we are stimulating the release of dynorphin (perhaps mimicking some aspects of withdrawal), levels of met-enkephalin increase in response to this, as a homeostatic mechanism to relieve this withdrawal-like state. This release profile is unique to the vNAcSh suggesting that this region is critical in the regulation of hedonic homeostasis. Though we have previously shown that dyn mRNA expression is similar in the vNAcSh and dNAcSh (*Al-Hasani et al., 2015*), it would be ideal to also verify the expression pattern and distribution of the enkephalins in the vNAcSh and dNAcSh. Finally, it should be noted that this detection technique maintains the poor temporal resolution of peptide and other microdialysis approaches. Therefore, conclusions regarding the differential transmitter and peptide release profiles should be tempered by the long time course of collection.

The ability to detect multiple peptides and small molecules in vivo with high sensitivity, as shown here demonstrates the potential of this method. It is, however, important to acknowledge limitations and future considerations with this methodology. Due to the structural similarity between ME/LE and dyn, we included DYN* as our only internal standard. An ideal assay would include isotopically labeled internal standards for each analyte of interest, however, the addition of analytes to detect in the ion trap reduces points per peak and complicates the chromatogram.

In this current method the 3 peaks of interest, LE/ME/dyn, all appear in the chromatogram in close proximity between 4–5.5 min, which may not seem ideal. A way to resolve this would be to flatten out the solvent gradient to better resolve the peaks chromatographically, however the length of the run would be longer, therefore reducing overall throughput. Here it takes 18 min from injection-to-injection and it is important to note that the peaks, while not fully resolved in time, were fully resolved by mass-to-charge ratio, so the separation was sufficient in this case. It is important to consider this delicate balance going forward to detect more peptides in an extended experimental paradigm and for concurrent detection of other peptides within individual samples.

Here, we only detected dyn1-8, LE and ME, but as mentioned in the introduction there are many more fragments that could be involved or altered that we have not measured. We primarily focused on dyn1-8 as this was the fragment that was the most reliable to detect in developing this methodology. Since we were at the early stages of method development it was crucial to limit variability where possible. The limitations to in vivo detection of neuropeptides by microdialysis with LC-MS has been thoroughly explored and are largely due to absorption and lack of adequate recovery (*Zhou et al., 2015*). In this current method, organic modifiers are not added to the perfusate for dialysis or to the sample in the vial, as we validated that there was no peptide carryover/stickiness in the LC-MS lines without such agents. However, it has been shown that the addition of an organic modifier, like acetonitrile, to a sample could be optimized for each peptide fragment and improve quantitation by reducing stickiness to the vial and LC-MS fluidic pathway (*Zhou et al., 2015*). This approach, however, appears to be dependent on each individual peptide.

This new approach, we describe here allows for detection of cell-type-specific evoked in vivo neuropeptide release within neural circuits to be observed during freely moving behavior. Neuropeptide biology remains a challenging territory in neuroscience. Methods to detect in vivo release of endogenous peptides are limited, preventing the full understanding of their properties and dynamics. The method described here is a first step in closing this gap in our understanding of neuropeptide-containing circuits during behavior.

## Materials and methods

### Key resources table

| Reagent type (species) or resource | Designation | Source or reference | Identifiers |
|---|---|---|---|
| Antibody | Neurotrace | Invitrogen | RRID:SCR_008410 |
| Peptide, recombinant protein | Dynorphin A 1–8 (dyn) | Bachem 4005845 | |

*Continued on next page*

Continued

| Reagent type (species) or resource | Designation | Source or reference | Identifiers |
|---|---|---|---|
| Peptide, recombinant protein | Leu-Enkephalin (LE) | Bachem 4006097 | |
| Peptide, recombinant protein | Met-Enkephalin (ME) | Sigma M6638 | |
| Chemical compound, drug | Vectashield | Vector Labs | |
| Software, algorithm | Thermo Xcalibur QuanBrowser | ThermoFisher | RRID:SCR_008452 |
| Other | pAAV-EF1α-double floxed-hChR2(H134R)-eYFP-WPRE-HGHpA | Addgene | RRID:SCR_002037 |
| Other | Agilent 1100 HPLC pump | Agilent Technologies | RRID:SCR_013575 |
| Other | linear ion trap | LTQ XL, Thermo Scientific | RRID:SCR_014992 |
| Other | Accela UHPLC system/TSQ Quantum Ultra triple quadrupole mass spectrometer | ThermoFisher | RRID:SCR_008452 |

## Viral preparation

Plasmids encoding pAAV-EF1α-DIO-eYFP [final titer $5 \times 10^{12}$ vg/ml], pAAV-EF1α-double floxed-hChR2(H134R)-eYFP-WPRE-HGHpA [final titer $2 \times 10^{13}$ vg/ml], were obtained from Addgene (AddgeneRRID:SCR_002037) originally from the Deisseroth Laboratory at Stanford University. The DNA was amplified with a Maxiprep kit (Promega) and packaged into AAV5 serotyped viruses by the WUSTL Hope Center Viral Core.

## Animals and surgical procedure

Adult male C57BL/6 mice (Envigo, 5–6 weeks of age) were used for initial experiments to determine perfusion media conditions and effects of probe design. For optogenetic studies, adult male prepro-dynorphin-IRES-cre (dyn-Cre) (RRID:IMSR_JAX:027958) mice were used. Mice were unilaterally injected with 300 nL of AAV5-EF1α-DIO-ChR2-eYFP with cre recombinase targeting to the Pdyn locus (Washington University in St. Louis, Hope Center Viral Vector Core, viral titer $2 \times 10^{13}$ vg/mL) into either the dNAcSh or vNAcSh and were allowed to recover from surgery at Washington University in St. Louis 1 week prior to shipment. Mice were then shipped to, and acclimated for 2–3 weeks at, the University of Michigan before probe implantation.

## Chemicals

Dynorphin A$_{1-8}$ (abbreviated dyn) and Leu-Enkephalin (LE) were purchased from Bachem (4005845 and 4006097, Torrance, CA); Met-Enkephalin (ME) was purchased from Sigma Aldrich (M6638, St. Louis, MO). Isotopically labeled leucine ($^{13}C_6{}^{15}N_1$-leucine) was used to create an isotopically labeled dynorphin A $_{1-8}$internal standard (DYN*) through the University of Michigan's protein synthesis core. Water, methanol, and acetonitrile for mobile phases are Burdick and Jackson HPLC grade purchased from VWR (Radnor, PA). All other chemicals were purchased from Sigma Aldrich (St. Louis, MO) unless otherwise noted. Artificial cerebrospinal fluid (aCSF) consisted of 145 mM NaCl, 2.67 mM KCl, 1.4 mM CaCl$_2$, 1.01 mM MgSO$_4$, 1.55 mM Na$_2$HPO$_4$, and 0.45 mM Na$_2$H$_2$PO$_4$ adjusted to pH 7.4 with NaOH. Ringer solution consisted of 148 mM NaCl, 2.7 mM KCl, 2.4 mM CaCl$_2$, and 0.85 mM MgCl$_2$ adjusted pH to 7.4 with NaOH. In experiments that used high K$^+$ ringer solution NaCl was adjusted to 48 mM and KCl was adjusted to 100 mM, all other chemicals remained the same.

## Fabrication of opto-dialysis probe

Fabrication of the optogenetic fiber optic probe was made as previously described (*Al-Hasani et al., 2015*; *McCall et al., 2015*; *Sparta et al., 2011*). Fabrication of the microdialysis probe and opto-dialysis probe are described in the supplemental information.

## In vivo microdialysis

Mice were group housed in temperature and humidity-controlled rooms with 12-hr light/dark cycles with access to food and water *ad libitum*. Both the Washington University in St. Louis and University of Michigan Unit for Laboratory Animal Medicine approved animal procedures, and they were in accordance with the National Institute of Health Guidelines for the Care and Use of Laboratory Animals. All experiments were conducted within the guidelines of Animal Research Reporting in vivo Experiments. Surgical procedures for inserting probes were similar to those previously described (*Mabrouk et al., 2011*; *Al-Hasani et al., 2015*; *McCall et al., 2015*; *Patterson et al., 2015*). Briefly, mice were anesthetized in an induction chamber with 5% isoflurane prior to surgical procedures and placed in a Model 963 stereotaxic frame (David Kopf Instruments, Tujunga, CA, USA) equipped with a mouse ear and bite bar. Mice were maintained under anesthesia with 1–2% isoflurane during cannulation procedures. A custom-made 1 mm polyacrylonitrile membrane (Hospal AN69) concentric probe, was inserted into either the dNAcSh (stereotaxic coordinates from bregma:+1.3 anterior-posterior [AP],±0.5 medial-lateral [ML], −4.5 mm dorsal-ventral [DV] or vNAcSh (stereotaxic coordinates from bregma:+1.3 [AP],±0.5 [ML], −5.0 mm [DV]). Light power from a 473 nm laser was measured at the membrane to ensure satisfactory light power (defined as ≥5 mW at a distance of 1 mm from the end of fiber optic) before implantation of microdialysis probes integrated with fiber optic (opto-dialysis probe). Implanted probes were secured using two bone screws and dental cement. Mice were allowed to recover 24 hr with free access to food and water prior to baseline collection for microdialysis studies.

For microdialysis studies, the fiber optic was connected to the laser via a tether running through a Ratturn (Bioanalytical Systems, Inc.) along with microdialysis perfusion lines. Microdialysis probes were flushed for 1 hr using a Fusion 400 syringe pump (Chemyx, Stafford, TX USA) at a flow rate of 2 µL/min. The flow rate was lowered to 0.8 µL/min and flushed for an additional 1 hr prior to fraction collection. Microdialysis fractions were collected every 15 min, resulting in a 12 µL sample. 2 µL of sample was removed for BzCl derivatization (*Lam et al., 2010*) to monitor small molecules, and the remaining 10 µL was spiked with 1.1 µL of 100 pM isotopically labeled DYN* (10 pM final concentration) and was used for peptide analysis.

When experiments were completed, mice were euthanized and perfused with paraformaldehyde. Brains were extracted to confirm probe placement and virus expression by histology. Mice with verified virus expression and correct probe placement were included in the data set.

## Optogenetic stimulation

On the day of the experiment, mice were connected to a laser via a tether alongside the microdialysis perfusion lines. An Arduino UNO was programmed and connected to the 473 nm laser to provide stimulation frequency of 10 Hz, 10 ms pulse width. The laser was manually operated and turned on and off during a single 15 min fraction after three baseline collections. Six additional fractions were collected after the photostimulation, followed by two fractions with high 100 mM $K^+$ ringer solution for a total of 12 fractions.

## Peptide assays with nanoflow LC-MS

An assay was developed to monitor opioid peptides (dyn, LE, and ME) using nanoflow LC-MS. Capillary columns and electrospray ionization emitter tips were prepared in-house (*Mabrouk et al., 2011*; *Patterson et al., 2015*). Capillary columns were prepared using a 10 cm length of 50/360 µm (inner diameter/outer diameter) fused silica capillary packed with 5 µm Altima™ C18 particles to a bed length of 3.5 cm. The column was connected to a fused silica ESI emitter tip using a Teflon connector.

5 µL samples were injected onto the capillary column. An Agilent 1100 HPLC pump (Agilent Technologies, RRID:SCR_013575, Santa Clara, CA) was used to deliver the elution gradient containing water with 0.1% FA for mobile phase A and mobile phase B was MeOH with 0.1% FA delivered as initial 0% B; 1 min, 30 %B; 4 min, 50% B; 4.1 min 100% B; 7 min, 100% B; 7.1 min, 0%B; and 10 min, 0% B. The capillary column was interfaced to a linear ion trap (LTQ XL, Thermo Scientific, RRID:SCR_014992), operating in positive mode. The MS (*Zhou et al., 2015*) pathway for opioid peptides dyn, LE, and ME were detected using *m/z* values of 491 → 435, 556 → 397, and 574 → 397 respectively. Isotopically labeled DYN* (+7 mass shift, isotopically labeled $^{13}C_6{}^{15}N_1$-leucine) internal standard was

detected using *m/z* values of 495 → 438. In vitro recovery of a 1 mm probe for dyn, LE, and ME were 12 ± 2%, 13 ± 3%, and 13 ± 2%.

Fresh dyn, LE, and ME standards, spiked with DYN* for the opioid assay were prepared daily. Standards were analyzed with nLC-MS at 0.01, 0.05, 0.1, 1, 10, 20, 50, and 100 pM concentrations in triplicate to determine linearity, reproducibility, and limits of detection. Opioid analytes were normalized to DYN*. Limits of detection for dyn, LE, and ME were 0.2 ± 0.04, 0.5 ± 0.3, and 0.6 ± 0.4 pM, respectively, in 5 µL and were determined each day of experimentation. Average carry over across all experiments for dyn, LE, and ME were 0.8 ± 0.03%, 0.3 ± 0.3%, and 0.3 ± 0.2% determined by running the highest calibration point immediately followed by a blank and integrating the peak area across the same time. Mice that had average basal levels above the limits of detection and had appropriate probe placement and virus expression were used for the study.

### Small molecule analysis using benzoyl chloride (BzCl) LC-MS

For small molecule analysis, dialysate samples were derivatized with BzCl and analyzed by LC-MS (*Song et al., 2012*; *Wong et al., 2016*). This BzCl assay targeted dopamine, GABA and glutamate. 2 µL dialysate were aliquoted from the peptide samples and were derivatized with 1.5 µL sodium carbonate, 100 mM; 1.5 µL BzCl, 2% (*v/v*) BzCl in acetonitrile; 1.5 µL isotopically labeled internal standard mixture diluted in 50% (*v/v*) acetonitrile containing 1% (*v/v*) sulfuric acid and spiked with deuterated ACh and Ch (C/D/N isotopes, Pointe-Claire, Canada). Derivatized samples were analyzed using Thermo Fisher Accela UHPLC system interfaced to a Thermo Fisher TSQ Quantum Ultra triple quadrupole mass spectrometer (Thermo Fisher Scientific RRID:SCR_008452) fitted with a HESI II ESI probe, operating in multiple reaction monitoring. 5 µL samples were injected onto a Phenomenex core-shell biphenyl Kinetex HPLC column (2.1 mm x 100 mm). Mobile phase A was 10 mM ammonium formate with 0.15% formic acid, and mobile phase B was acetonitrile. The mobile phase was delivered an elution gradient at 450 µL/min as follows: initial, 0% B; 0.01 min, 19% B; 1 min, 26% B; 1.5 min, 75% B; 2.5 min, 100% B; 3 min, 100% B; 3.1 min, 5% B; and 3.5 min, 5% B. Thermo Xcalibur QuanBrowser (Thermo Fisher Scientific RRID:SCR_008452) was used to automatically process and integrate peaks. Each peak was visually inspected to ensure proper integration.

### Assessment of probe placement

Mice were anesthetized with pentobarbital and transcardially perfused with ice-cold 4% paraformaldehyde in phosphate buffer (PB). Brains were dissected, post-fixed for 24 hr at 4°C and cryoprotected with solution of 30% sucrose in 0.1M PB at 4°C for at least 24 hr, cut into 30 µm sections and processed for Nissl body staining. Sections were washed three times in PBS and blocked in PBS containing 0.5% Triton X-100 (G-Biosciences) for 1 hr. This was followed by a 1 hr incubation with fluorescent Nissl stain to allow visualization of cell bodies (1:400, Neurotrace, Invitrogen RRID:SCR_008410). Sections were then washed three times in PBS, followed by three 10 min rinses in PB and mounted on glass slides with Hard set Vectashield (Vector Labs) for episcope microscopy. Correct regional expression of the AAV5-DIO-ChR2-eYFP was verified in addition to placement of the opto-dialysis probe in either the vNAcSh or dNAcSh, which are represented on hit maps.

### Statistical analyses

The University of Michigan Center for Statistical Consultation and Research helped design a linear mixed model analysis appropriate for this study using SPSS Statistics software (SPSS, RRID:SCR_002865). The linear mixed model analysis was chosen to account for variations within and between mice and to account for missing data points within individual animals following sample loss or mechanical failure of the instrument. The linear mixed model was used to determine differences in basal conditions, effect of photostimulation relative to basal conditions, and prolonged effects after photostimulation relative to basal conditions. Linear mixed models were used to compare between genotypes within each region sampled, and between regions within genotypes. In vitro data were represented as mean ± SD and the in vivo was represented as mean ± SEM. In all cases significance was defined as p≤0.05.

## Acknowledgements

This work is supported by NIDA R01 DA033396 (MRB), NIDA K99 DA038725 (RA), NIMH F31 MH101956 (JGM), WUSTL DBBS (JGM) and NIBIB R01 EB003320 (RTK). We thank the Bruchas Laboratory and Kennedy Laboratory for valuations discussions and support. We especially thank Shanna Resendez and Curtis Austin from the Aragona Laboratory for helpful discussion and technical assistance. We thank Karl Deisseroth (Stanford) for the channelrhodopsin-2 (H134) construct, Bradford Lowell (Harvard) and Michael Krashes (NIDDK) for the preprodynorphin-IRES-cre mice. We also thank The WUSTL HOPE Center viral vector core (NINDS, P30NS057105) and the University of Michigan Center for Statistical Consultation and Research (CSCAR, consultant Yumeng Li) and Luke Ziolkowski from the McCall Laboratory at Washington University who assisted with statistical analysis.

## Additional information

### Competing interests

Michael R Bruchas: Co-founder of Neurolux, Inc, a company that is making wireless optogenetic probes. None of the work in this manuscript used these devices or is related to any of the company's activities, but we list this information here in full disclosure. The other authors declare that no competing interests exist.

### Funding

| Funder | Grant reference number | Author |
| --- | --- | --- |
| National Institute on Drug Abuse | K99/R00 Pathway to Independence DA038725 | Ream Al-Hasani |
| National Institute of Mental Health | F31 MH101956 | Jordan G McCall |
| National Institute of Biomedical Imaging and Bioengineering | R01 EB003320 | Robert T Kennedy |
| National Institute on Drug Abuse | R01 DA033396 | Michael R Bruchas |

The funders had no role in study design, data collection and interpretation, or the decision to submit the work for publication.

### Author contributions

Ream Al-Hasani, Robert T Kennedy, Conceptualization, Resources, Data curation, Formal analysis, Supervision, Funding acquisition, Validation, Investigation, Visualization, Methodology, Writing—original draft, Project administration, Writing—review and editing; Jenny-Marie T Wong, Conceptualization, Data curation, Formal analysis, Supervision, Validation, Investigation, Visualization, Methodology, Writing—original draft, Project administration, Writing—review and editing; Omar S Mabrouk, Conceptualization, Data curation, Formal analysis, Supervision, Validation, Investigation, Visualization, Methodology, Writing—original draft, Project administration; Jordan G McCall, Data curation, Validation, Investigation, Visualization, Methodology, Writing—original draft, Project administration; Gavin P Schmitz, Validation, Investigation, Visualization; Kirsten A Porter-Stransky, Investigation; Brandon J Aragona, Conceptualization; Michael R Bruchas, Conceptualization, Resources, Data curation, Supervision, Funding acquisition, Validation, Investigation, Visualization, Methodology, Writing—original draft, Project administration, Writing—review and editing

### Author ORCIDs

Ream Al-Hasani http://orcid.org/0000-0002-8781-6234
Jordan G McCall http://orcid.org/0000-0001-8295-0664
Michael R Bruchas http://orcid.org/0000-0003-4713-7816

## Ethics

Animal experimentation: This study was performed in strict accordance with the recommendations in the Guide for the Care and Use of Laboratory Animals of the National Institutes of Health. All of the animals were handled according to approved institutional animal care and use committee (IACUC) protocols of Washington University in St. Louis and the University of Michigan.

## Decision letter and Author response

Decision letter https://doi.org/10.7554/eLife.36520.013
Author response https://doi.org/10.7554/eLife.36520.014

# Additional files

## Supplementary files

• Supplementary file 1. Opto-dialysis probe fabrication
DOI: https://doi.org/10.7554/eLife.36520.009

• Transparent reporting form
DOI: https://doi.org/10.7554/eLife.36520.010

## Data availability

All data generated or analysed during this study are included in the manuscript and supporting files.

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
