## [Decision Letter]

Thank you for submitting your article "in vivo detection of optically-evoked opioid peptide release" for consideration by *eLife*. Your article has been reviewed by three peer reviewers, including Julie A Kauer as the Reviewing Editor and Reviewer #1, and the evaluation has been overseen by a Senior Editor. The following individuals involved in review of your submission have agreed to reveal their identity: Christopher Evans (Reviewer #2); Richard E Mains (Reviewer #3).

The reviewers have discussed the reviews with one another and the Reviewing Editor has drafted this decision to help you prepare a revised submission. Given the nature of this contribution, we feel it would best be considered as a Tools and Resources paper where the novelty and potential application of the technique is given paramount consideration. The Short Report format that you adopted for your submission suits this purpose well so the difference will be in title only with little distinction in the manner in which your paper would be posted.

Summary:

This paper develops a new technique for opioid peptide detection in brain in vivo. The microdialysis probe used to collect samples is directly attached to a fiber optic to allow optogenetic activation of specific cell types. The authors demonstrate that by driving dynorphin-containing neurons, they can detect different levels of dynorphin and other neurotransmitters in distinct parts of the nucleus accumbens.

Essential revisions:

The reviewers thought that the paper makes a significant contribution to the field, but would like elaboration on several points. The paper presents a novel methodological approach, but more scientific discussion would enhance the presentation.

Specific points for revision:

1) Some thoughtful explanations are needed and some caveats re statements about some of the findings (e.g. about Figure 3D):

For example looking at the data for Met-Enk release (Figure 3D) given the variability and approximately equivalent mean, it is difficult to make the statement – "detectable increases in ME were observed during photostimulation of dyn-containing cells in the vNAcSh (interaction effect; t =2.824, p=0.006) and not the dNAcSh)". Some additional analysis should be conducted to assess if there is differences between the regions in this measure.

I don't understand the base line change in Figure 3C vNAcSh in fraction 4 (optical stimulation fraction) – this would change the conclusions of Leu-Enk-stimulated release in this area. Whether the Leu-Enk measured is derived from pEnk or pDyn is an important question (given opioid receptor selectivity and a δ-Mu agonist released from pDyn cells).

2) The data showing changes in Leu-Enk (and trend for/significant Met-Enk release) following optogenetic of Dyn-containing cells in dorsal and ventral NAc shell could implicate that the Leu Enk is derived from pDyn. However, there are several alternative explanations, including that the Leu-Enk derived from p-Dyn may be swamped by pEnk derived Leu-Enk. Inferences could be made using the pEnk KO mice – perhaps future experiments.

3) It is difficult to easily understand the release of Met-Enk with optical stimulation of PDyn in the NAc. The MSN pDyn D1 neurons project to the VTA so I am thinking collaterals but the neurotransmitter is GABA (and pDyn of course) which should be inhibitory on Met-Enk containing D2 neurons. How there is stimulated DA release is also a bit of a puzzle since the collaterals are GABAergic. Miss-expression of the virus in cholinergic interneurons could explain some of the results – hence the request for specificity of ChR expression. The study does assume that expression of ChR2 is discrete and only in pDyn-expressing neurons and some concrete evidence of expression would be important for this paper.

4) Absolute level ranges in the dialysate should be calculable given the internal standard – ranges are important to be reported given this is predominantly a methodological paper.

5) An important improvement would be to spread out the liquid chromatography step (Figure 1C) to make resolution and subsequent quantification better – one simply needs to flatten out the solvent gradient a touch to get better. The authors do not need to do this in their figure, but they should discuss/incorporate this technical point as a suggestion future users should consider.

*Reviewer #1:*

This is primarily a methods paper, but the apposition of microdialysis and optogenetic drive is an elegant addition to the tools available, and the authors do a nice job of presenting a little scientific data along with their methodological advance.

There is no discussion of how or why dopamine is being released upon dyn-cre optogenetic stimulation. This is an important result for interpretation of numerous results in the field and for interpretation of dyn-cre data going forward. Since presumably dyn-cre cells in accumbens release dynorphin and GABA, how do the authors think this works? Through projections to the VTA? These results affect past work using these mice beyond al-Hasani et al 2015, and this would be worth some discussion.

Reviewer #2:

The authors describe the development of a probe combining optogenetic stimulation and microdialysis for nLC-MS detection of endogenous peptide release in vivo. The regulation of neuropeptide release is an important and very difficult area of research. All elements of the technology proposed have been widely applied previously (dialysis for peptides, nLC-MS for peptide dialysate detection and optogenetics for region-specific stimulated release of neuropeptides). However, the combinational probe and its utility for the prodynorphin system (a notoriously difficult system to assess by dialysis because of dynorphins being "sticky" peptides) and in the mouse that has a very small brain, provide innovative aspects. The authors use this methodology in transgenic mice to photostimulate dynorphin cells in the ventral NAcSh to elicit robust aversive behavior and in the dorsal NAcSh to induce place preference with different peptide dialysate profiles attributed.

1) A sentence or two should be added regarding the opioid system and selectivity of Dyn 1-8 and Met/Leu enkephalin for different opioid receptors and which precursors these peptides can be derived (pEnk or pDyn). Also, there are many other different kappa-selective products from pDyn such as α and β neoendorphin, Dyn 1-17 and Dyn B and some discussion as to why they were not measured in this study is warranted.

2) Whether the Leu-Enk measured is derived from pEnk or pDyn is an important question (given opioid receptor selectivity and a δ-Mu agonist released from pDyn cells). The data showing changes in Leu-Enk (and trend for/significant Met-Enk release) following optogenetic of Dyn-containing cells in dorsal and ventral NAc shell could implicate that the Leu Enk is derived from pDyn. However, there are several alternative explanations, including that the Leu-Enk derived from p-Dyn may be swamped by pEnk derived Leu-Enk. Inferences could be made using the pEnk KO mice – perhaps future experiments.

3) The inclusion of a peptide isomer standard is important. One problem is sticking of standard peptides to apparatus for analysis (tubing, columns etc.) so the fact that the analysis apparatus is peptide-unseen is important since wash-off with dialysate can give false positive readings. It was not quite clear when the isotopically labeled Dyn 1-8 was introduced to the sample (before or after the dialysis) and an experimental timeline could be added to help understand the protocol. Was the dialysis probe and MS analysis apparatus always naïve to standard peptides?

3) A little puzzling is why there is trending increased K-evoked release of peptide reserves (Figure 3—figure supplement 1Figure 2) in the Cre vs. non Cre animals since depletion might be anticipated after optical stimulation.

4) In Figure 2 the scale of 5mm does not look correct given a 1mm dialysis membrane and Figure 2A could be better resolution.

*Reviewer #3:*

This brief manuscript describes the development of an approach to couple microdialysis and high-sensitivity mass spec assays with optogenetic (blue light) stimulation of depolarization in cells expressing the Dynorphin family of peptides. Results then look at peptide release and also secretion of several conventional neurotransmitters. The paper is concise and for the most part well explained, with the exceptions below.

The use of an internal isotopically-labeled standard (DYN*) is a really good idea, modeled after similar approaches in other fields. The comment (Discussion, second paragraph) that more internal standards would be ideal but too costly is not really appropriate, for this reviewer, since the punchline of Figure 1 is that the method is reproducible and capable of reliable measurements. All the standard synthetic peptides have a Tyr residue, hence a good A280, so concentrations are easy to determine. Having more isotopically labeled standards would simply clutter up the chromatograms.

The AAV-DIO-ChR2-eYFP (also called AAV5.…) needs to be described and presumably accompanied by a literature citation. This construct presumably has a floxed STOP codon in front of the ChR2, which is either a fusion protein with eYFP or uses an IRES. The readers and reviewers should not be left guessing and presuming.

The infusion of 100 mM K^+^ (Results, third paragraph) similarly needs more explanation and description. Is this isotonic? Buffered? What other ingredients, since 100 mM is quite hypotonic?

The experimental description (Results, third paragraph) states that mice were included in the data if K^+^ stimulated secretion and if correct anatomical probe placement occurred and if viral expression were confirmed = 3 ways to fail. The text should give some indication of the failure rate for each – are we looking at 10% of mice passing all three tests or 75% or what?

---

## [Author Response]

1) Some thoughtful explanations are needed and some caveats re statements about some of the findings (e.g. about Figure 3D):For example looking at the data for Met-Enk release (Figure 3D) given the variability and approximately equivalent mean, it is difficult to make the statement – "detectable increases in ME were observed during photostimulation of dyn-containing cells in the vNAcSh (interaction effect; t =2.824, p=0.006) and not the dNAcSh)". Some additional analysis should be conducted to assess if there is differences between the regions in this measure.

We thank the reviewers for bringing these key points to our attention. We have added further details and statistics to explain the observed changes in Met-Enk (ME) in the vNAcSh compared to the dNAcSh and below:

“Converse to this, we observed a significant increase in ME during photostimulation of dyn-containing cells in the vNAcSh (comparing cre+ and cre, interaction effect; t =2.824, p=0.006). […] However, the lack of a significance between Cre+ and Cre- in dNAcSh is likely due, at least in part, to the fact the baseline levels in dNAcSh are higher than the vNAcSh (7.114 pM versus 2.71 pM, respectively).”

I don't understand the base line change in Figure 3C vNAcSh in fraction 4 (optical stimulation fraction) – this would change the conclusions of Leu-Enk-stimulated release in this area. Whether the Leu-Enk measured is derived from pEnk or pDyn is an important question (given opioid receptor selectivity and a δ-Mu agonist released from pDyn cells).

This is an important comment. The key take home for us was that we are able to measure three different opioid peptides in a single sample being release from these Dyn+, D1+ neurons. Specifically, we wanted to ensure that we were able to detect dynorphin and leu-enk without cross talk between the two, which is indeed the case here, partly due to the release profiles being very different. We absolutely agree that the drift in baseline may affect the interpretations and conclusions, but this is often seen in dialysis studies over time, and this was a first of its kind approach. Furthermore, we have shown here that photostimulation can change peptide concentration, which was in line with our primary goal. Using this current method it is not possible to know whether LE is derived from pDyn or pENK but we plan in future studies to combine this with conditional knockout approaches to selectively delete the peptides (and we added a statement to the Discussion to address this point) in specific cell types, so that we can later determine what happens to these signals in the absence of particular peptides in D1 vs. D2 neurons in the region. We have also included this discussion – information in the first paragraph of the Discussion. In addition, we cannot preclude interactions between MSN cell types, specifically that optogenetic activation of D1 cells could impart changes on D2 cell activity thereby releasing Met-Enk, indirectly, and/or via indirect circuits outside the NAc.

2) The data showing changes in Leu-Enk (and trend for/significant Met-Enk release) following optogenetic of Dyn-containing cells in dorsal and ventral NAc shell could implicate that the Leu Enk is derived from pDyn. However, there are several alternative explanations, including that the Leu-Enk derived from p-Dyn may be swamped by pEnk derived Leu-Enk. Inferences could be made using the pEnk KO mice – perhaps future experiments.

Yes, these are excellent points. We have addressed in the reviewer comment and response above.

3) It is difficult to easily understand the release of Met-Enk with optical stimulation of PDyn in the NAc. The MSN pDyn D1 neurons project to the VTA so I am thinking collaterals but the neurotransmitter is GABA (and pDyn of course) which should be inhibitory on Met-Enk containing D2 neurons. How there is stimulated DA release is also a bit of a puzzle since the collaterals are GABAergic. Miss-expression of the virus in cholinergic interneurons could explain some of the results – hence the request for specificity of ChR expression. The study does assume that expression of ChR2 is discrete and only in pDyn-expressing neurons and some concrete evidence of expression would be important for this paper.

We thank the reviewers for these thoughtful insights. See above, in comment above about possible interactions between MSN microcircuits. Furthermore, dopamine release via D1 MSN stimulation, could be via two mechanisms 1) via indirect suppression of D1 projections back to VTA GABA neurons (see Edwards et al., 2017, from Bonci’s group) which acts via GABA-B to disinhibit VTA DA neurons. This would then result in dopamine release in the NAc following D1 stimulation. Furthermore, an alternative, presynaptic regulation of DA release from VTA to NAc projections is also plausible, and recently been investigated by multiple groups including Lemos et al., 2016, and Dobbs et al., 2016 (Discussion, first paragraph).

To address selectivity of our labeling approach, we attempted an immunohistochemistry experiment in which we stained tissue expressing AAV5-EF1α-DIO-ChR2-eYFP in dynorphin cells in the ventral nucleus accumbens with two different ChAT antibodies from two different species. We were hoping that these stains would show that dynorphin cells expressing ChR2 are distinct from ChAT labeled cells. Unfortunately, we were unable to see antibody staining and robust expression despite our multiple attempts using a rabbit and goat raised ChAT antibody. Please see Author response image 1 showing ChR2 expression and nissl staining. We were disappointed that this was not more successful but there is some anecdotal evidence suggesting that these antibodies can often be unreliable and success rates differ between lots and laboratories.

However, we think this is a minor concern, given having previously shown using antibodies and in situ hybridization assays that ChR2 is in fact localized to neurons expressing dynorphin and dopamine receptors (D1) which are unique to medium spiny neurons and not tonically active ChAT neurons (Al-Hasani et al., 2015). Furthermore, in this paper we go on to show extensively that CHR2 expression using this same virus is localized to Dynorphin Cre+ neurons, and effect at producing Kappa-opioid dependent behavioral effects. Thus, the impact of ChAT neuron expression of ChR2 in non D1 cells is very unlikely, given our prior published results, as well as those of other collaborative groups using the same virus and mouse cre-driver line (Crowley et al., Cell Reports, 2016).

4) Absolute level ranges in the dialysate should be calculable given the internal standard – ranges are important to be reported given this is predominantly a methodological paper.

This is a useful addition. The absolute concentrations are reported in the text:

“Though data is represented as% baseline the range of absolute dialysate concentration detected for dyn was 0.28-0.44 pM in dNAcSh and 0.13-0.28 pM in vNAcSh; LE 1.39-3.28 pM in dNAcSh and 1.30- 2.24 pM in vNAcSh; ME 6.19-8.69 pM in dNAcSh and 2.57-4.11 pM.”

5) An important improvement would be to spread out the liquid chromatography step (Figure 1C) to make resolution and subsequent quantification better – one simply needs to flatten out the solvent gradient a touch to get better. The authors do not need to do this in their figure, but they should discuss/incorporate this technical point as a suggestion future users should consider.

This is an insightful comment and we have added this to the discussion of methodological limitations going forward. This can be found in the fourth paragraph of the Discussion.

Reviewer #1:

This is primarily a methods paper, but the apposition of microdialysis and optogenetic drive is an elegant addition to the tools available, and the authors do a nice job of presenting a little scientific data along with their methodological advance.There is no discussion of how or why dopamine is being released upon dyn-cre optogenetic stimulation. This is an important result for interpretation of numerous results in the field and for interpretation of dyn-cre data going forward. Since presumably dyn-cre cells in accumbens release dynorphin and GABA, how do the authors think this works? Through projections to the VTA? These results affect past work using these mice beyond al-Hasani et al 2015, and this would be worth some discussion.

We thank the reviewer for this insightful comment, we have added discussion relevant to this point above, and in the Discussion as well.

Reviewer #2:

[…] 1) A sentence or two should be added regarding the opioid system and selectivity of Dyn 1-8 and Met/Leu enkephalin for different opioid receptors and which precursors these peptides can be derived (pEnk or pDyn). Also, there are many other different kappa-selective products from pDyn such as α and β neoendorphin, Dyn 1-17 and Dyn B and some discussion as to why they were not measured in this study is warranted.

We appreciate this comment. Additional background information about the opioid system and peptide selectivity has now been added to the Introduction (third paragraph). Further discussion about the limitations of detection due to absorption and recovery has also been added to the fifth paragraph of the Discussion.

2) Whether the Leu-Enk measured is derived from pEnk or pDyn is an important question (given opioid receptor selectivity and a δ-Mu agonist released from pDyn cells). The data showing changes in Leu-Enk (and trend for/significant Met-Enk release) following optogenetic of Dyn-containing cells in dorsal and ventral NAc shell could implicate that the Leu Enk is derived from pDyn. However, there are several alternative explanations, including that the Leu-Enk derived from p-Dyn may be swamped by pEnk derived Leu-Enk. Inferences could be made using the pEnk KO mice – perhaps future experiments.

Discussed above in more detail.

3) The inclusion of a peptide isomer standard is important. One problem is sticking of standard peptides to apparatus for analysis (tubing, columns etc.) so the fact that the analysis apparatus is peptide-unseen is important since wash-off with dialysate can give false positive readings. It was not quite clear when the isotopically labeled Dyn 1-8 was introduced to the sample (before or after the dialysis) and an experimental timeline could be added to help understand the protocol. Was the dialysis probe and MS analysis apparatus always naïve to standard peptides?

This is an important consideration. The isotope was added after dialysis, so the probe is naïve to the standard. This information is included in the second paragraph of the subsection “In vivo microdialysis”. We have also added more information to the timeline in Figure 3 to help clarify the methodology.

We have also included the following statement in the text to address your concern:

“Limits of detection for Dyn, LE, and ME were 0.2 ± 0.04, 0.5 ± 0.3, and 0.6 ± 0.4 pM in 5 μL determined each day of experimentation. Average carry over across all experiments are now also included for dyn, LE, and ME were 0.8 ± 0.03%, 0.3 ± 0.3%, and 0.3 ± 0.2% determined by running the highest calibration point immediately followed by a blank and integrating the peak area across the same time.”

3) A little puzzling is why there is trending increased K-evoked release of peptide reserves (Figure 3—figure supplement 1) in the Cre vs. non Cre animals since depletion might be anticipated after optical stimulation.

This is an interesting point however we cannot be sure that optical stimulation does in fact deplete peptides to the point that this could be a detectable difference. Further, given that firing patterns of neurons elicited via optogenetic stimulation may not in fact mimic the stochastic asynchronous nature of MSN firing in vivo. This is a clear limitation of optogenetics in general in this approach, and we have now included a statement in the Discussion to address this point. Furthermore, K^+^ likely has much more diffuse targeting, spreading further and activating more cells than photostimulation resulting in larger gross release in both genotypes.

4) In Figure 2 the scale of 5mm does not look correct given a 1mm dialysis membrane and Figure 2A could be better resolution.

The scale appears correct but to clarify this we have added a 1mm scale bar to the right hand image.

Reviewer #3:

[…] The use of an internal isotopically-labeled standard (DYN*) is a really good idea, modeled after similar approaches in other fields. The comment (Discussion, second paragraph) that more internal standards would be ideal but too costly is not really appropriate, for this reviewer, since the punchline of Figure 1 is that the method is reproducible and capable of reliable measurements. All the standard synthetic peptides have a Tyr residue, hence a good A280, so concentrations are easy to determine. Having more isotopically labeled standards would simply clutter up the chromatograms.The AAV-DIO-ChR2-eYFP (also called AAV5.…) needs to be described and presumably accompanied by a literature citation. This construct presumably has a floxed STOP codon in front of the ChR2, which is either a fusion protein with eYFP or uses an IRES. The readers and reviewers should not be left guessing and presuming.

This has been corrected to AAV5-EF1α-DIO-ChR2-eYFP and updated throughout the text.

The infusion of 100 mM K^+^ (Results, third paragraph) similarly needs more explanation and description. Is this isotonic? Buffered? What other ingredients, since 100 mM is quite hypotonic?

We have ensured that all this information is contained within the Materials and methods, subsection “Chemicals”. K^+^ is in the same aCSF medium used for the rest of the experiment however K^+^ was increased to 100mM and Na^+^ reduced to 48mM to account to changes in osmolarity.

The experimental description (Results, third paragraph) states that mice were included in the data if K^+^ stimulated secretion and if correct anatomical probe placement occurred and if viral expression were confirmed = 3 ways to fail. The text should give some indication of the failure rate for each – are we looking at 10% of mice passing all three tests or 75% or what?

81.1% (15 out of 18) of mice passed all three tests. This information can be found in the third paragraph of the Results.